# Ribosome•RelA structures reveal the mechanism of stringent response activation

**Anna B Loveland**[1,2,3,4]**, Eugene Bah**[1,2†]**, Rohini Madireddy**[1,2]**, Ying Zhang**[1,2]**, Axel F Brilot**[3,4‡§]**, Nikolaus Grigorieff**[3,4,5*]**, Andrei A Korostelev**[1,2*]

[1]RNA Therapeutics Institute, University of Massachusetts Medical School, Worcester, United States; [2]Department of Biochemistry and Molecular Pharmacology, University of Massachusetts Medical School, Worcester, United States; [3]Department of Biochemistry, Brandeis University, Waltham, United States; [4]Rosenstiel Basic Medical Sciences Research Center, Brandeis University, Waltham, United States; [5]Janelia Research Campus, Howard Hughes Medical Institute, Ashburn, United States

**Abstract** Stringent response is a conserved bacterial stress response underlying virulence and antibiotic resistance. RelA/SpoT-homolog proteins synthesize transcriptional modulators (p)ppGpp, allowing bacteria to adapt to stress. RelA is activated during amino-acid starvation, when cognate deacyl-tRNA binds to the ribosomal A (aminoacyl-tRNA) site. We report four cryo-EM structures of *E. coli* RelA bound to the 70S ribosome, in the absence and presence of deacyl-tRNA accommodating in the 30S A site. The boomerang-shaped RelA with a wingspan of more than 100 Å wraps around the A/R (30S A-site/RelA-bound) tRNA. The CCA end of the A/R tRNA pins the central TGS domain against the 30S subunit, presenting the (p)ppGpp-synthetase domain near the 30S spur. The ribosome and A/R tRNA are captured in three conformations, revealing hitherto elusive states of tRNA engagement with the ribosomal decoding center. Decoding-center rearrangements are coupled with the step-wise 30S-subunit 'closure', providing insights into the dynamics of high-fidelity tRNA decoding.

**\*For correspondence:** niko@ grigorieff.org (NG); andrei. korostelev@umassmed.edu (AAK)

**Present address:** †Mayo Medical School, Rochester, United States; ‡Department of Biochemistry and Biophysics, University of California, San Francisco, United States; §School of Medicine, University of California, San Francisco, United States

## Introduction

RelA/SpoT homolog (RSH) proteins play a central role in bacterial stringent response—a major stress-response pathway and key driver of bacterial virulence and antibiotic resistance (*Neidhardt, 1987*; *Gentry et al., 2000*; *Pizarro-Cerda and Tedin, 2004*; *Dalebroux et al., 2010a*; *Gao et al., 2010*; *Nguyen et al., 2011*; *Dordel et al., 2014*). In response to environmental stress, such as nutrient deprivation, RSH proteins synthesize small-molecule 'alarmones' collectively referred to as (p)ppGpp (i.e., guanosine pentaphosphate and guanosine tetraphosphate; [*Cashel and Gallant, 1969*] and reviewed in [*Potrykus and Cashel, 2008a*; *Atkinson et al., 2011*]). Accumulation of (p)ppGpp activates transcription of genes required for stress response, inhibits transcription of genes required for replication and growth, and reformats the transcription of metabolic genes according to the stress condition (*Polakis et al., 1973*; *Mittenhuber, 2001*; *Magnusson et al., 2005*; *Jain et al., 2006a*; *Kuroda, 2006*; *Wang et al., 2007*; *Ferullo and Lovett, 2008*; *Potrykus and Cashel, 2008a*; *Traxler et al., 2008*; *Dalebroux et al., 2010a*; *Dalebroux et al., 2010b*). Inactivation of RSH proteins in pathogenic bacteria dramatically reduces pathogenicity and bacterial load in the host, up to ~10,000-fold for *Salmonella* Typhimurium (*Na et al., 2006*;

*Sun et al., 2009*; *Dalebroux et al., 2010a*; *Vogt et al., 2011*). Understanding the molecular mechanism of RSH activation may therefore guide the development of new antibacterial therapeutics.

In *E. coli*, RelA synthesizes alarmones in response to amino acid deprivation (*Cashel and Gallant, 1969*; *Haseltine et al., 1972*). When the supply of amino acids becomes limiting, binding of cognate but uncharged (deacylated) transfer RNA (tRNA) to the A (aminoacyl-tRNA) site of the 70S ribosome activates RelA (*Haseltine and Block, 1973*; *Richter, 1976*; *Wendrich et al., 2002*). The 70S•RelA•deacyl-tRNA complex triggers RelA to transfer a pyrophosphoryl group from ATP to GTP or to GDP to form pppGpp or ppGpp, respectively (*Haseltine and Block, 1973*; *Sy and Lipmann, 1973*; *Wendrich et al., 2002*). RelA also binds to the ribosome in the absence of deacyl-tRNA, but this binding does not stimulate (p)ppGpp synthesis (*Haseltine and Block, 1973*; *Ramagopal and Davis, 1974*; *Wagner and Kurland, 1980*; *Wendrich et al., 2002*).

RelA, a 744 amino-acid protein (~84 kDa), consists of functionally distinct halves. The N-terminal half (amino acids 1–380) includes a pseudo-hydrolase (inactive-hydrolase) domain (aa 1–200) and the (p)ppGpp synthetase domain (aa ~201–380). A crystal structure of the N-terminal half of the *Streptococcus equisimilii* RelA homolog RelSeq (aa 1–385) showed that the synthetase domain belongs to the nucleotidyltransferase superfamily and identified the catalytic residues (*Hogg et al., 2004*). The C-terminal half of RelA (aa ~400–744) and other RSH proteins is thought to control the synthetase activity of the N-terminal region (*Schreiber et al., 1991*; *Gropp et al., 2001*; *Yang and Ishiguro, 2001a*; *Mechold et al., 2002*; *Avarbock et al., 2005*; *Jain et al., 2006b*). Dimerization (*Yang and Ishiguro, 2001a*) or oligomerization (*Gropp et al., 2001*; *Avarbock et al., 2005*; *Jain et al., 2006b*) of free (ribosome-unbound) RelA and RSH proteins are thought to contribute to negative regulation of the synthetase activity. A 10.8-Å cryo-EM reconstruction of a 70S•RelA•deacyl-tRNA complex revealed a bi-lobed density overlapping with the elongation-factor-binding site near the A site (*Agirrezabala et al., 2013*). The position of RelA and conformation of deacyl-tRNA resembled those of elongation factor EF-Tu and aminoacyl tRNA in the 70S•EF-Tu•aa-tRNA pre-accommodation-like complexes (*Stark et al., 2002*; *Valle, 2002*; *Schmeing et al., 2009*). However, the resolution of the map did not allow model building, leaving the molecular details of RelA activation unclear (*Agirrezabala et al., 2013*). A lack of high-resolution structures of full-length RelA or its homologs bound to the ribosome precludes our ability to understand the regulation of RelA synthetase activity.

To understand how RelA activates stringent response on ribosomes bound with cognate deacyl-tRNA, we sought a high-resolution structure of the entire 70S•RelA•deacyl-tRNA complex. Single-particle electron cryo-microscopy (cryo-EM) and maximum-likelihood classification of a single dataset yielded four different cryo-EM structures of *E. coli* RelA bound to the *E. coli* ribosome at 3.9-Å to 4.1-Å resolution. As in cryo-EM structures of other ribosome complexes (e.g. [*Greber et al., 2014*; *Fischer et al., 2015*]), the local resolution of our maps in the ribosome core is higher than the average resolution of the maps, allowing for structural interpretation in central regions at near-atomic resolution. The structures reveal large-scale conformational rearrangements in RelA when it binds deacyl-tRNA entering the 30S A site, suggesting a mechanism of activation of the (p)ppGpp synthetase. Furthermore, distinct conformations of the deacyl-tRNA, 30S subunit, and the ribosomal decoding center reveal structural dynamics of tRNA binding in the presence of RelA and suggest why stringent response activation requires cognate tRNA.

## Results and discussion

We used single-particle cryo-EM to obtain the structures of full-length *E. coli* RelA bound to *E. coli* 70S•tRNA ribosome complex programmed with an mRNA coding for tRNA^fMet in the P site and tRNA^Phe in the A site. Maximum-likelihood classification using FREALIGN (*Lyumkis et al., 2013*) revealed four unique classes of ribosome particles containing RelA (*Figure 1A–D*, *Figure 1—figure supplements 1* and *2*, *Figure 1—source data 1*). In all RelA-bound structures, the ribosome contains P-site and E-site tRNAs and adopts the classical, non-rotated conformation (*Cate et al., 1999*; *Frank and Agrawal, 2000*; *Yusupov et al., 2001*), similar to that of the post-translocation-ribosome with peptidyl-tRNA in the P site (*Voorhees et al., 2009*; *Jenner et al., 2010*). In Structure I, the ribosome A site is vacant, and RelA is bound via its C-terminal portion while its N-terminal domains are disordered (*Figure 1A*). In Structures II, III, and IV, the central and C-terminal parts of RelA are well resolved and the anticodon-stem loop (ASL) of a cognate deacyl-tRNA^Phe is bound to the A site of

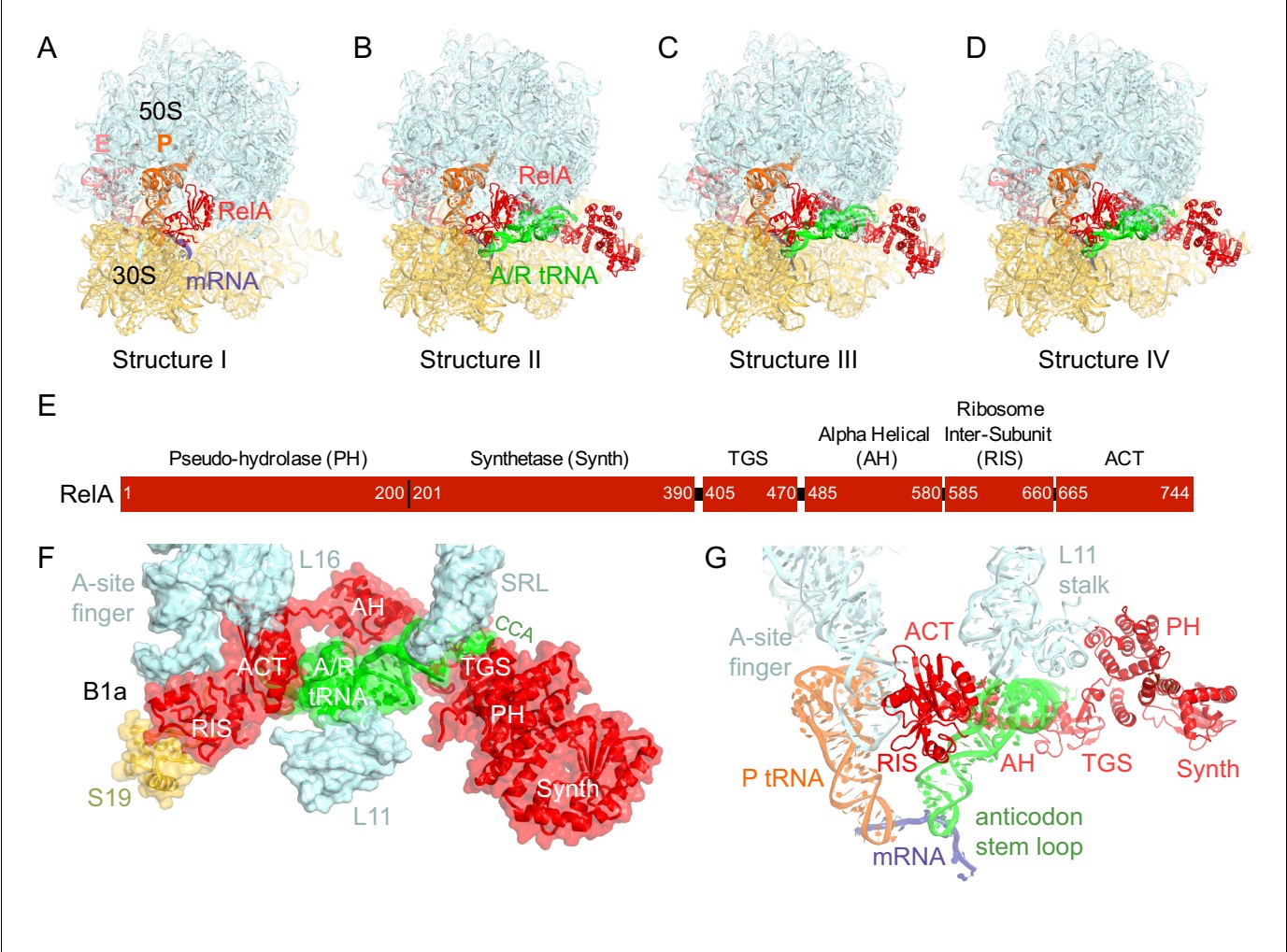

**Figure 1.** Cryo-EM structures of the 70S•RelA complexes. (**A**) Structure of the 70S•RelA complex lacking deacyl-tRNA in the A site (Structure I) reveals the C-terminal superdomain comprising the RIS and ACT domains (red). This superdomain binds near the A site at bridge B1a between the 30S and 50S subunits. (**B**) Structure of the 70S•RelA•deacyl-tRNA complex (Structure II) shows that the C-terminal superdomain is similar to that in Structure I (**A**). The central and N-terminal portions of the protein become visible upon interaction with A/R tRNA. (**C**, **D**) Structures III (**C**) and IV (**D**) are compositionally the same as Structure II, but differ in the conformations of A/R tRNA, the 30S subunit, and RelA. (**E**) Domain architecture of *E. coli* RelA. The numbers indicate amino acid positions in RelA. (**F**) RelA wraps around A/R tRNA. The model from Structure IV is shown in space-filling and secondary-structure rendering. Abbreviations used: ACT (*A*spartate kinase-*C*horismate mutase-*T*yrA domain), RIS (*R*ibosome-*I*nter*S*ubunit domain), AH (α-helical domain), TGS (ThrRS, GTPase, SpoT/RelA domain), Synth (synthetase domain), PH (pseudo-hydrolase domain), SRL (sarcin-ricin loop), B1a (bridge B1a between S19 and A-site finger), CCA (three 3'-terminal nucleotides of tRNA). (**G**) The position of RelA is shown relative to the A/R tRNA, P-site tRNA and mRNA in Structure IV. 16S rRNA and ribosomal proteins are omitted for clarity. In all panels, the 50S subunit is colored pale blue; the 30S subunit, yellow; RelA, red; A/R tRNA, green; P-site tRNA, orange; E-site tRNA, pink; and mRNA, dark blue.

The following source data and figure supplements are available for figure 1:

**Source data 1.** Structure I-IV map resolution and refinement statistics.

**Figure supplement 1.** Schematic of cryo-EM refinement and classification procedures.

**Figure supplement 2.** Cryo-EM density in Structures I-IV.

**Figure supplement 3.** Interactions of the L11 stalk with A/R tRNA.

**Figure supplement 4.** Domain organization of RelA.

the 30S subunit (*Figure 1B–D*). We refer to the deacyl-tRNA bound to the 30S A site and RelA as A/R (A/RelA) tRNA. Structures II, III, and IV differ from each other in the conformations of RelA, A/R tRNA, the 30S subunit, and the L11 stalk of the 50S subunit. The N-terminus of ribosomal protein L11 is required for RelA activation in 70S•RelA•deacyl-tRNA complexes (*Friesen et al., 1974*; *Parker et al., 1976*; *Yang and Ishiguro, 2001b*; *Wendrich et al., 2002*; *Jenvert and Holmberg Schiavone, 2007*; *Shyp et al., 2012*). The L11 N-terminus interacts with A/R tRNA but not with RelA in Structures II, III, and IV (*Figure 1F–G*, *Figure 1—figure supplement 3*). Thus, the lack of (p) ppGpp synthetase activity of RelA on mutant ribosomes missing L11 or the L11 N-terminus (*Friesen et al., 1974*; *Wendrich et al., 2002*; *Jenvert and Holmberg Schiavone, 2007*) is likely due to the inability of these ribosomes to coordinate deacyl-tRNA for activation of RelA.

## The C-terminal domain of RelA binds at intersubunit bridge B1a

Although activation of RelA catalysis requires cognate, deacylated A-site tRNA (*Haseltine and Block, 1973*), RelA can bind to the ribosomes in the absence of A-site tRNA (*Ramagopal and Davis, 1974*; *Richter et al., 1975*; *Richter, 1976*; *Wendrich et al., 2002*). Our map lacking A-site tRNA (Structure I) reveals density for the C-terminal region of RelA (aa ~530–744) near the A and P sites (*Figure 1—figure supplement 2A*). Specifically, the C-terminal region binds the intersubunit bridge B1a (*Yusupov et al., 2001*), which connects the A-site finger (helix 38) of the large subunit to the head of the small subunit (*Figures 1E–G* and *2A*). The lack of density for the central and N-terminal domains of RelA suggests that these regions are not ordered in the absence of deacylated A-site tRNA (*Figure 1—figure supplement 2A*).

The resolved C-terminal region comprises two domains. The ACT domain (aspartate kinase-chorismate mutase-tyrA; residues 665–744; [*Atkinson et al., 2011*]) is composed of four β-strands covered by two α-helices (*Grant, 2006*), and lies in the 50S subunit within a cavity formed by the A-site finger, helix 89, L16 and P-site tRNA (*Figure 2A–B* and *Figure 2—figure supplement 1*). The tips of two β-hairpins, including Q705-Q706 and highly conserved D675, interact with R50 and R51 of L16 (*Figure 2B*). The β-strands of ACT form a platform that packs against the 3′-strand of the A-site finger. Here, the N-terminal and the C-terminal β-strands of ACT, including R670 and R739, interact with the bulged A896 of the A-site finger helix (*Figure 2B* and *Figure 2—figure supplement 1A*). The fold of the ACT domain was previously noted to resemble the RNP motif that binds single-stranded RNA, but no RNA-binding ACT domains had been identified (*Burd and Dreyfuss, 1994*; *Grant, 2006*). Our findings reveal that while the RelA ACT domain is not sequence-homologous to RNP motifs, it interacts with double-stranded RNA via the same face that RNP domains use to bind single-stranded RNA.

The second binding interface between the C-terminal region and the ribosome involves a previously unclassified RelA domain (aa 585–660). The domain bridges the large and the small ribosomal subunits (*Figures 1E–F*, *2A–B* and *Figure 2—figure supplement 1B*). We refer to this domain as RIS (*R*ibosome-*I*nter*S*ubunit) domain. The domain core contains a four-stranded β-sheet and a short α-helix and is structurally similar to a zinc-finger domain (*Lee et al., 1989*). On the 50S subunit, the α-helix (residues 638–647) docks into the minor groove of the A-site finger and interacts with 23S rRNA residues 883–885 and 892–894 (*Figure 2A–B*). On the 30S subunit, the β-sheet of RIS packs at the hydrophobic patch of the β-sheet of S19, comprising V57, P58 and F60.

In summary, Structure I reveals that the ACT and RIS domains of RelA form a C-terminal superdomain that anchors RelA to the 70S ribosome, consistent with reduced binding of RelA to ribosomes upon mutation or deletion of the C-terminal domain (*Yang and Ishiguro, 2001a*). The density for the RIS domain core is well resolved in Structure I (*Figure 1—figure supplement 2A*), supporting the model that amino acids encompassing the RIS domain form the major ribosomal-binding domain of RelA (*Yang and Ishiguro, 2001a*).

## Deacyl-tRNA pins the TGS domain against the 30S subunit, exposing the dynamic N-terminal domains near the spur

Structures II, III, and IV contain RelA bound to the ribosome with cognate deacyl-tRNA in the A site of the 30S subunit (*Figure 1B–D*). The ribosome structures share an overall conformation, including the relative positions of ribosomal subunits, tRNAs and RelA. As described in the earlier cryo-EM study of RelA ribosome complexes (*Agirrezabala et al., 2013*), the positions of A/R tRNA in the

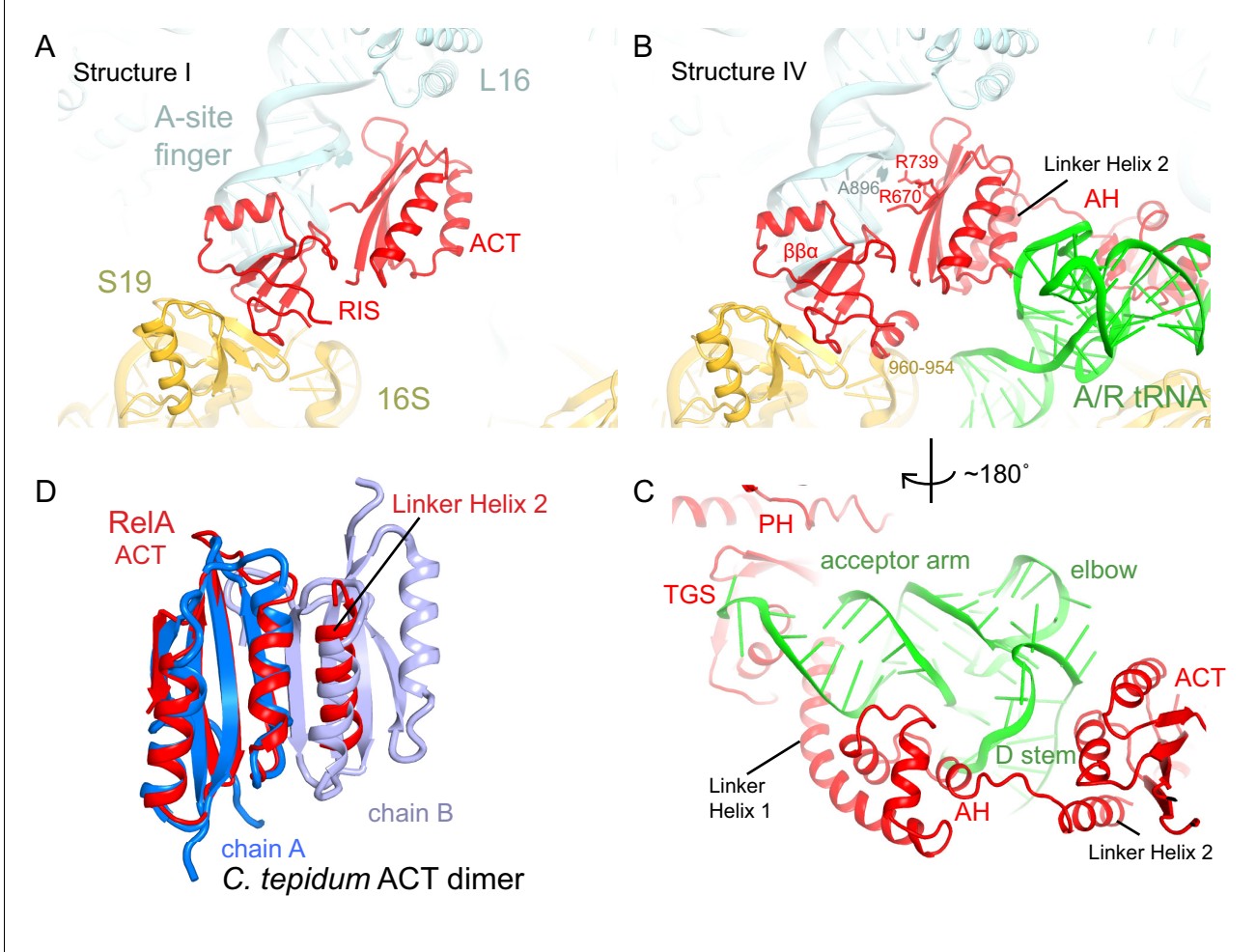

**Figure 2.** The C-terminal superdomain of RelA binds at the intersubunit bridge B1a. (**A**) In the absence of A-site tRNA (Structure I), the C-terminal superdomain of RelA interacts with the intersubunit bridge B1a. The ACT domain interacts with the A-site finger (helix 38 of 23S rRNA) and L16. The RIS domain interacts with the A-site finger, 16S rRNA and S19. (**B**) In the presence of A/R tRNA, as in Structures II, III or IV (shown), the RIS and ACT domains maintain their interaction with the ribosome at bridge B1a, and additional elements of the C-terminal region become ordered. Linker Helix 2, which connects the α-helical (AH) and RIS domains, packs against the ACT domain (also shown in (**C**)), similarly to an α-helix in the isolated ACT dimer shown in (**D**). (**C**) In Structures II, III, and IV (shown), the AH domain of RelA interacts with the D stem and acceptor arm of A/R tRNA. The AH domain is connected to the TGS domain by Linker Helix 1 that passes under the acceptor arm of A/R tRNA. (**D**) Comparison of the ACT domain in the 70S-bound RelA and in the solution structure of the isolated ACT dimer of *C. tepidum* RSH (PDB: 2KO1, [*Eletsky et al., 2009*]). In the 70S•RelA structures (Structure III is shown), Linker Helix 2 is positioned similarly to a helix from the partner ACT molecule (light blue) in the dimerized ACT domain. The interaction between the linker helix and the ACT domain in Structures II, III and IV suggests that the possible dimerization surface of the ACT domain in free RelA is disrupted upon ordering of RelA by the deacyl-tRNA in the A/R conformation. The colors in all panels are as in *Figure 1*.

The following figure supplements are available for figure 2:

**Figure supplement 1.** Cryo-EM density for the RIS, ACT and AH domains.

**Figure supplement 2.** Re-evaluation of the previously reported 10.8 Å cryo-EM map.

RelA-bound structures globally resemble that of the A/T pre-accommodated aminoacyl-tRNA captured in the ribosome in the presence of EF-Tu (*Stark et al., 2002*; *Valle, 2002*; *Schmeing et al., 2009*). However, we observe important differences between the A/R and A/T tRNAs, and among ribosome conformations, as discussed in a following section.

Structures II, III, and IV reveal density for the entire RelA protein when cognate deacyl-tRNA is bound in the 30S A site. In these structures, RelA wraps around the tRNA and adopts a boomerang-

like conformation (*Figure 1F*). This RelA conformation contrasts with the compact RelA conformation occupying the elongation-factor binding site that was proposed based on the 10.8-Å map of a 70S•RelA•deacyl-tRNA complex (*Agirrezabala et al., 2013*). However, our re-evaluation of the lower-resolution map revealed previously unassigned density near the intersubunit bridge B1a, which makes that map consistent with the extended RelA conformation shown here (*Figure 2—figure supplement 2*).

In Structures II, III and IV, the C-terminal RIS and ACT domains form one wing of the boomerang-like structure at bridge B1a and are positioned similarly to those in Structure I (*Figure 2A–B*). The N-terminal part of the RIS domain (at aa 590–595) is also stabilized by interactions with the phosphate backbone of G987 and the 954–960 loop of the 16S ribosomal RNA (rRNA), which forms a wall of the A-site tRNA-binding cavity. The C-terminal domains are connected with the central α-helical domain (AH; aa 485–580; *Figure 2B–C*). The core of the AH domain (aa ~520–560) is formed by short helices, which pack near the D stem of the A/R tRNA. The ~15-amino-acid C-terminal helix of the AH domain connects the AH core with the RIS domain (Linker helix 2; *Figure 2B–C*). Linker helix 2 packs against the ACT domain similarly to an α-helix of the dimerization partner of the isolated ACT domain of *Chlorobium tepidum* RSH (PDB: 2KO1; [*Eletsky et al., 2009*]) (*Figure 2D*). This suggests that if the ACT domains were also dimerized in the full-length free RelA, the packing of this α-helical linker next to the ACT domain would be possible on the ribosome only upon dimer disassembly. The long N-terminal helix (Linker Helix 1), which lies under the acceptor arm of the A/R tRNA, connects the AH core with the ubiquitin-like TGS domain (amino acids 405–470; ThrRS, *G*TPase, and SpoT/RelA (*Sankaranarayanan et al., 1999*) (*Figure 2C*). Together the central AH and TGS domains form the elbow of the RelA boomerang. The TGS domain is pinned against 16S rRNA by the acceptor end of A/R tRNA (*Figure 3A–B*), as described below. The overall conformations of the TGS domain are similar between Structures II, III, and IV (*Figure 3—figure supplement 1A*). Finally, the N-terminal pseudo-hydrolase domain (aa 1–200) and synthetase domain (aa 200–380), forming the second wing of the boomerang, face the periphery of the ribosome in the vicinity of the 30S subunit spur (helix 6) (*Figure 4A*) and adopt a range of conformations in Structures II, III and IV, as discussed below.

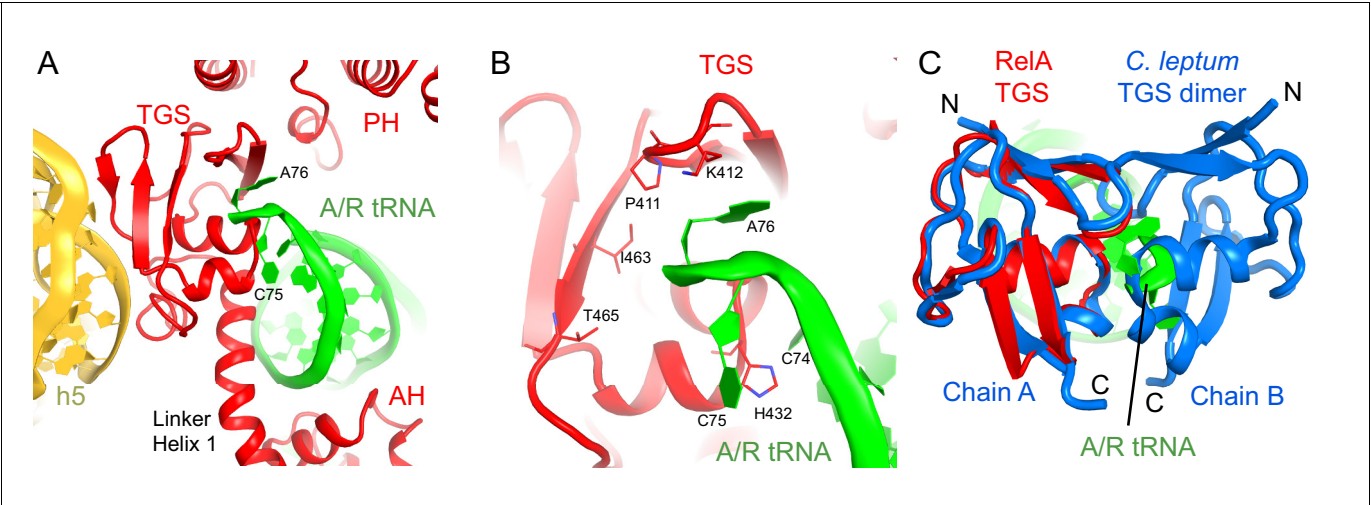

**Figure 3.** Interactions of the TGS domain of RelA with the A/R tRNA and 16S rRNA. (**A**) The 3′ CCA end of A/R tRNA pins the TGS domain against helix 5 of 16S rRNA. (**B**). Interactions of the terminal nucleotides of the A/R tRNA with the TGS domain. (**C**) Comparison of the TGS domain bound with the CCA end of the A/R tRNA (this work) with the dimeric TGS domain from *C. leptum* RSH (*Forouhar et al., 2009*), showing that the A/R tRNA disrupts the dimerization surface of the isolated homologous TGS domain. Superposition was performed by the structural alignment of the all-atom models of the TGS domain (Structure IV) and the TGS dimer (PDB: 3HVZ). The TGS dimer is shown in blue; other molecules are labeled and colored as in *Figure 1*. Structure IV is shown in all panels.

The following figure supplement is available for figure 3:

**Figure supplement 1.** Interactions of the TGS domain with the A/R tRNA.

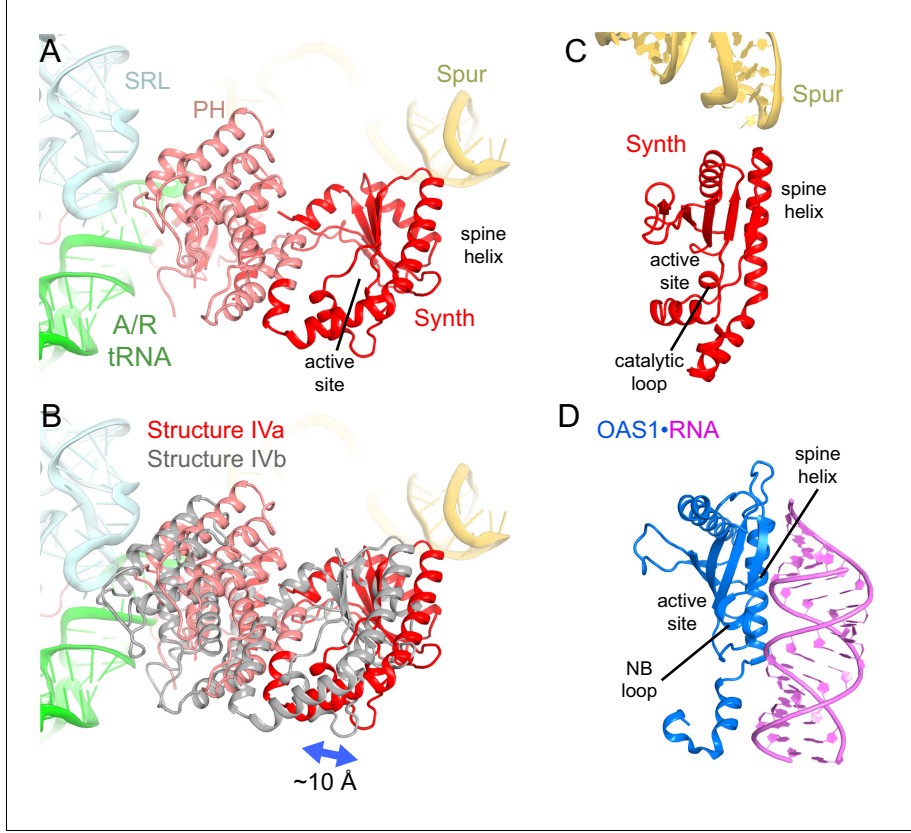

**Figure 4.** Positions and interactions of the N-terminal domains of RelA. (**A**) Pseudo-hydrolase (PH; pink) and synthetase (Synth; red) domains are in the intersubunit space between the sarcin-ricin loop (SRL) of the 23S rRNA and the spur of the 16S rRNA. The N-terminal domains are shown in a conformation, in which the synthetase domain is near the spur (Structure IV is shown). (**B**) Comparison of the two conformations of the N-terminal domains inferred from the heterogeneous cryo-EM density by additional sub-classification (Structure IV is shown; see also *Figure 4—figure supplement 1*). The red model is shown as in (**A**). The gray model exhibits a conformation shifted away from the spur. (**C**) Relative positions of the synthetase domain and the spur in Structure IVa. (**D**) Structure of the innate immune sensor OAS1 (blue, PDB: 4RWP) bound with an RNA helix (magenta) (*Lohöfener et al., 2015*). OAS1 is a second-messenger-(2'-5'-oligoadenylate)-synthesizing enzyme, whose architecture resembles that of the synthetase domain of RelA, shown in a similar orientation in (**C**). The nucleotide-binding loop (NB loop) and other structural elements are labeled.

The following figure supplements are available for figure 4:

**Figure supplement 1.** Cryo-EM densities for the N-terminal domains, obtained by sub-classification of Structures II, III and IV.

**Figure supplement 2.** Comparison of the synthetase domain of RelA with metazoan innate immune sensors OAS1 and cGAS.

The TGS domain interacts with the 3′ CCA end of deacylated A/R tRNA and with 16S rRNA helix 5 (*Figure 3A*). The $^{74}CCA^{76}$ end adopts a conformation similar to that in the EF-Tu-bound A/T-tRNA, in which C75 is bulged out (*Schmeing et al., 2009*; *Fischer et al., 2015*), whereas C74 and A76 interact with several conserved residues of the protein. H432 stabilizes the CCA conformation by intercalating between C74 and C75 (*Figure 3B*). P411 and K412 on a β-hairpin loop interact with the nucleobase of A76 (*Figure 3B*). The ribose of A76 faces a β-sheet at residues 461–465 (*Figure 3B*). (p)ppGpp synthesis by RelA requires deacyl-tRNA binding to the ribosome (*Haseltine and Block, 1973*). An amino acid bound at A76 would sterically clash with the β-sheet of TGS and prevent binding of A/R tRNA to the TGS domain, in keeping with the inability of charged

tRNA to activate RelA. To gain insights into the role of interaction between the CCA end and the TGS domain, we compared our structures with the X-ray structure of the isolated TGS domain from *C. leptum* RSH (PDB: 3HVZ) (*Forouhar et al., 2009*). The isolated TGS domain forms a dimer. The comparison reveals that the A/R tRNA disrupts the dimerization interface (*Figure 3C*). Since oligomerization of RSH proteins inhibits (p)ppGpp production (*Gropp et al., 2001*; *Yang and Ishiguro, 2001a*; *Avarbock et al., 2005*; *Jain et al., 2006b*), interaction with deacyl-tRNA may contribute to RelA activation *via* stabilization of the monomeric RelA.

The N-terminal pseudo-hydrolase and synthetase domains in the intersubunit space of the ribosome were poorly resolved in the maps for Structures II, III and IV, suggesting structural heterogeneity. To resolve this heterogeneity, we performed data sub-classification, using a focused spherical mask (*Grigorieff, 2016*) covering the N-terminal region of RelA, separately for Structures II, III and IV. Sub-classification of each structure into three or more classes revealed two predominant classes that resolved the shapes of the N-terminal region, which differ in position by at least 10 Å (*Figure 4—figure supplement 1A–F*). Classification into seven classes reveals the appearance of less resolved density between these two extreme conformations, suggesting that the N-terminal domain samples a continuum of positions (*Figure 4—figure supplement 1G*). The maps of the two predominant conformations allowed rigid-body fitting of the homology model of the *E. coli* RelA pseudo-hydrolase and synthetase domains, obtained from the crystal structure of the isolated N-terminal domain from *S. equisimilis* RelSeq (*Hogg et al., 2004*). In the first conformation, the synthetase domain is positioned near the spur. The synthetase domain contains a long helix (the spine helix, at aa 208–243), which buttresses the catalytic region at the β-platform (aa 248–340; *Figure 4A and C*) (*Hogg et al., 2004*). The loop, which connects the spine helix with the β-platform (at residues 244–246) approaches the tip of the 16S spur within ~5 Å, suggesting that the protein interacts with rRNA, but the details of the possible interaction cannot be visualized in the low-resolution maps (*Figure 4—figure supplement 1A–C*). In the second conformation, the synthetase domain is separated from the spur by shifting away from its first predominant conformation by ~10 Å (*Figure 4B*). The pseudo-hydrolase domain is bound near the sarcin-ricin loop of the large subunit (nt 2653–2667 of 23S rRNA). The homologous RelSeq contains a functional hydrolase domain, and allosteric regulation was proposed to govern the switch between (p)ppGpp hydrolase and synthetase activities (*Hogg et al., 2004*). It is possible that activation of the synthetase domain of RelA involves conformational rearrangements between the pseudo-hydrolase and synthetase domains, triggered by movement relative to the TGS domain and/or interactions with ribosomal RNA. Alternatively, specific interactions of the synthetase domain with the spur may trigger the catalytic activity. The RelA synthetase domain structure resembles that of other second-messenger synthetases (*Figures 4D* and *Figure 4—figure supplement 2*), including: metazoan innate immune sensor OAS1, a 2′–5′–oligoadenylate synthase triggered by double-stranded RNA (*Donovan et al., 2013*; *Lohöfener et al., 2015*); metazoan cGAS, a cyclic-GMP–2′–5′–AMP synthase triggered by double-stranded DNA (*Civril et al., 2013*; *Gao et al., 2013*; *Kranzusch et al., 2013*; *Sun et al., 2013*); and *Vibrio cholerae* pathogenicity factor DncV, a cyclic-GMP–3′–5′–AMP synthase (*Kranzusch et al., 2014*). Innate immune

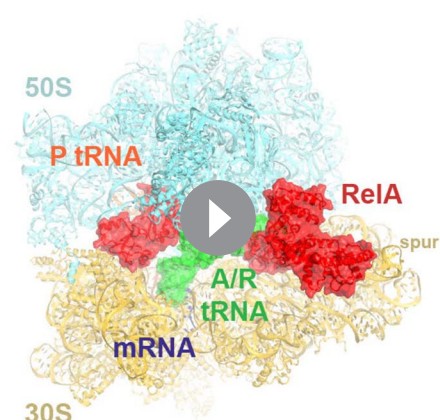

**Video 1.** An animation showing transitions between Structures I, II, III and IV. Three views (scenes) are shown: (1) A view of the complete 70S complex, as in *Figure 1*; two conformations of the N-terminal domain of RelA are shown for Structures II, III and IV. In Structure I, the central and N-terminal domains of RelA are not resolved – here, a model from Structure IV is shown in gray for reference. (2) A close-up view, showing A/R tRNA accommodation ('settling') into the decoding center and 30S domain rearrangements. The head of the 30S subunit is shown on the left, the body of the 30S is on the right, the shoulder is oriented toward the viewer. (3) A close-up view of the decoding center, similar to that shown in *Figure 6D–G*. Colors are as in *Figure 1*.

sensors OAS1 and cGAS are activated by binding of an RNA or DNA duplex, respectively, at the spine helix (*Civril et al., 2013*; *Donovan et al., 2013*; *Gao et al., 2013*; *Lohöfener et al., 2015*). The proximity of the spine helix of RelA to the spur highlights the possibility of activation of RelA *via* a mechanism reminiscent of those for OAS1 and cGAS innate immune sensors (*Figure 4—figure supplement 2*).

## Distinct intermediates of tRNA in the 30S A site of Structures II, III, and IV

Our structures reveal three conformations of the deacyl-tRNA and the 30S decoding center, providing insights into the mechanism of specific RelA activation by cognate tRNA (shown as an animation in *Video 1* and http://labs.umassmed.edu/korostelevlab/msc/relamovie.gif). Activation of (p)ppGpp synthesis by RelA on the ribosome depends on the selection of tRNA cognate to the mRNA codon in the A site (*Haseltine and Block, 1973*). In their *in vitro* experiments, Haseltine and Block demonstrated that the substitution of cognate tRNA$^{Ala}$, which reads the GCA, GCC, GCG or GCU codons, with near-cognate tRNA$^{Val}$ (GUA, GUC, GUG, or GUU) or tRNA$^{Glu}$ (GAA or GAG) results in a more than 32-fold decrease in (p)ppGpp synthesis by RelA on ribosomes programmed with an Ala codon in the A site (*Haseltine and Block, 1973*). Our structures suggest stepwise accommodation of the codon-anticodon helix in the decoding center, which helps explain this exquisite sensitivity.

Classification of our cryo-EM data revealed three unique classes (Structures II, III, and IV) that differ in the conformation of A/R tRNA (*Figures 1B–D* and *Figure 1—figure supplement 1*). In all three structures, the A/R tRNA anticodon base-pairs with the mRNA codon in the 30S A site, the elbow contacts the L11 stalk, whereas the acceptor arm is located in the vicinity of the sarcin-ricin loop of 23S rRNA (*Figure 1F–G*). The A/R tRNAs are highly distorted relative to accommodated A-site tRNA, such that the anticodon-stem loop is kinked toward the CCA end of the tRNA (*Figure 5—figure supplement 1A*), somewhat similar to the A/T aminoacyl-tRNA in EF-Tu-bound pre-accommodation-like ribosome structures (*Stark et al., 2002*; *Valle, 2002*; *Schmeing et al., 2009*; *Fischer et al., 2015*). The A/R tRNAs, however, differ from the A/T tRNA in the degrees of twisting around residues 26 and 44, which link the anticodon-stem loop with the rest of tRNA (*Figure 5—figure supplement 1B*). The CCA end of A/R tRNA (Structure II) is positioned ~10 Å away from that in the A/T tRNA (*Figure 5—figure supplement 1B*). Relative to the P-site tRNA, the A/R elbows of all three RelA-bound structures are tilted by up to 10 Å farther than the A/T tRNA (*Figure 5—figure supplement 1A*). As such, the A/R tRNA appears even more slanted away from the ribosome core than the A/T tRNA.

Comparison of Structures II, III, and IV reveals a concerted movement of A/R tRNA and the RelA central domains toward the head of the 30S subunit, as if the tRNA gradually 'settles' into the A site of the 30S subunit from Structure II through III to IV (*Figure 5A–B*). The tRNA accommodation coincides with a conformational change in the 30S subunit termed 'domain closure' (*Ogle et al., 2001*; *Ogle et al., 2002*; *Jenner et al., 2010*; *Demeshkina et al., 2012*). The acceptor arm of A/R tRNA and the TGS domain of RelA shift toward the head of the 30S subunit by ~2 Å between Structures II and III, and by ~2 Å between Structures III and IV (*Figure 5C–D* and *Figure 6—figure supplement 1*). The shoulder of the 30S subunit also moves by nearly 5 Å toward the head and body from Structure II to IV (*Figure 6—figure supplement 1* and *Figure 6—source data 1*). In Structure II, the 30S subunit is in the open conformation observed previously in the absence of A-site tRNA (*Ogle et al., 2001*; *Jenner et al., 2010*), whereas in Structure IV, the 30S subunit is in the fully closed conformation (*Figure 6A–B* and *Figure 6—figure supplement 1*) (*Selmer et al., 2006*; *Jenner et al., 2010*; *Demeshkina et al., 2012*). The 30S subunit in Structure III adopts an intermediate state between the open and closed states (*Figures 6B* and *Figure 6—figure supplement 1*). Thus, Structure II represents a previously unseen open 30S subunit in the presence of the codon-anticodon interaction. Structure III, in turn, represents an intermediate in the 30S 'domain-closure' pathway.

The observation of open, intermediate, and closed conformations of the 30S subunit with A-site tRNA interacting with the mRNA codon prompted us to study the conformation of the decoding center in each structure in more detail. The local resolution of our maps in the decoding center is sufficient to determine nucleotide conformations (*Figure 6—figure supplements 2* and *3*). Studies of ribosome-tRNA complexes demonstrated that the decoding center plays a central role in cognate tRNA stabilization. Specifically, universally conserved nucleotides of the decoding center A1492, A1493, G530 of 16S rRNA and A1913 of 23S rRNA interact with the minor groove of the codon-

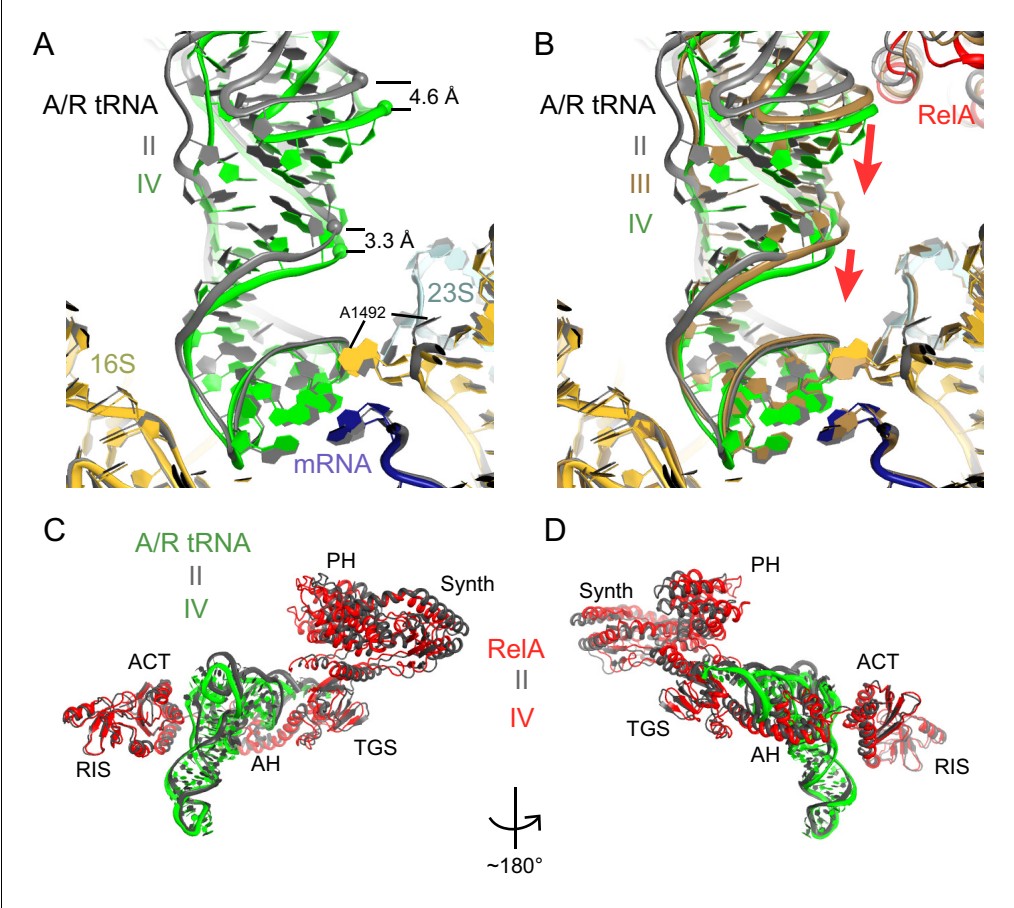

**Figure 5.** A/R tRNA and RelA rearrange toward the 30S subunit in Structures II to IV. (**A**) A/R tRNA settles into the decoding center of the 30S subunit between Structures II (grey) and IV (colored as in *Figure 1*). Structures II and IV were aligned on the 16S rRNA. RelA is not shown. The positions of A1492 in Structures II and IV are labeled for reference. (**B**) A/R tRNA and RelA positions in Structures II (grey), III (gold) and IV (colored as in *Figure 1*). (**C**) and (**D**) Two views showing that RelA shifts with the A/R tRNA between Structure II (grey) and Structure IV (colored as in *Figure 1*). The TGS domain, which interacts with the acceptor arm of A/R tRNA, moves more than the RIS and ACT domains. The superposition of Structures II and IV was performed by structural alignment of the 16S rRNA.

The following figure supplement is available for figure 5:

**Figure supplement 1.** Comparison of A/R tRNA to A/T and A/A tRNA.

anticodon interaction (*Ogle et al., 2001*; *Selmer et al., 2006*; *Jenner et al., 2010*). A1492 and A1493 are important in stabilizing Watson-Crick geometry of the first two base pairs (*Ogle et al., 2001*; *Demeshkina et al., 2012*), and are thus thought to contribute to tRNA recognition, providing high fidelity of protein synthesis (*Ogle et al., 2003*).

Despite the presence of an A-site tRNA in Structure II, the conformation of the decoding center resembles that found in the absence of an A-site tRNA with the 30S in an open conformation (compare *Figure 6E–D*) (*Ogle et al., 2001*; *Jenner et al., 2010*). Strong density shows that A1492 resides inside helix 44, as observed in the absence of an A-site tRNA (A1492 OFF) (*Ogle et al., 2001*; *Jenner et al., 2010*) (*Figures 6E* and *Figure 6—figure supplement 3B,F and I*). G530 is separated from A1492 by more than 10 Å (*Figure 6E* and *Figure 6—figure supplement 3B*), similar to that in the absence of an A-site tRNA (G530 in the OFF position). A1493 bulges out from helix 44 of 16S rRNA, so that the nucleotide is oriented toward the codon-anticodon helix. Weak density suggests that the base does not form a stable interaction with the codon-anticodon helix (*Figure 6—figure supplement 3I*), although A1493 appears pre-arranged for such an interaction by being bulged out

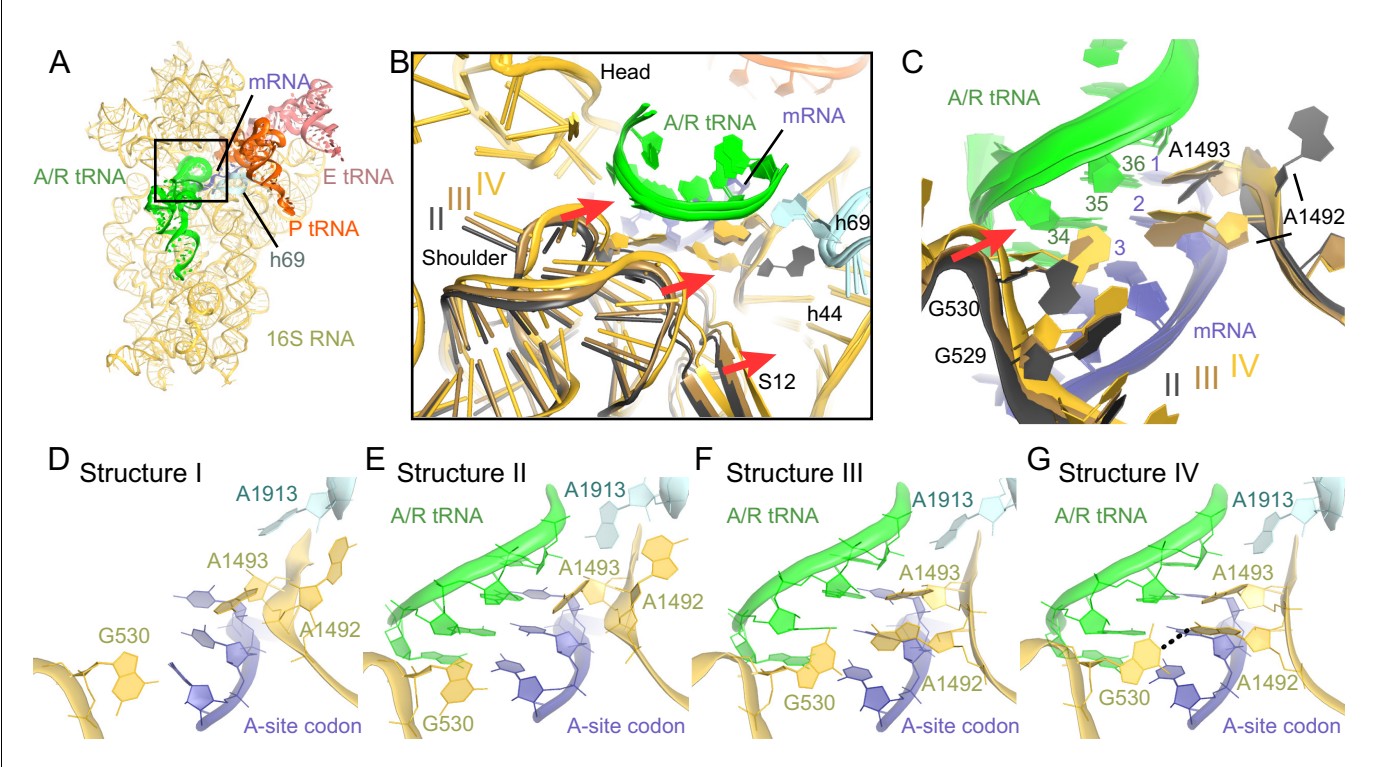

**Figure 6.** Closure of the 30S subunit and decoding-center rearrangements in Structures II, III and IV. (**A**) A view down on the 30S subunit from the inter-subunit interface shows the position of the decoding center (boxed). The 50S subunit (except for helix 69), small ribosomal proteins and RelA are omitted for clarity. (**B**) The conformational differences in the 30S subunits of Structures II, III and IV suggest a domain-closure pathway. From Structure II to IV, the 30S shoulder is shifted by more than 4 Å toward the 30S head. The superposition of Structures II, III, and IV was performed by structural alignment of nt 980–1200 of 16S rRNA, corresponding to the 30S head. (**C**) Conformational differences in the decoding-center's universally conserved nucleotides A1492, A1493, and G530 of Structure II, III, and IV are shown after alignment as in (**B**). (**D**) The decoding center of Structure I, which lacks A/R tRNA, is similar to that of the 30S domain-open structures with a vacant A site (*Ogle et al., 2001*; *Jenner et al., 2010*). (**E**) The decoding center of Structure II reveals a previously unseen state, in which the domain-open 30S subunit contains a tRNA in the A site. A1493 is near the first base pair of the codon-anticodon helix; A1492 is in helix 44, whereas G530 adopts the conformation previously observed in the absence of the A-site tRNA (*Ogle et al., 2001*; *Jenner et al., 2010*). (**F**) The decoding center of Structure III reveals a previously unseen state, in which the 30S subunit adopts an intermediate domain-closure conformation. A1493 and A1492 interact with the first and second base pairs of the codon-anticodon helix, respectively, whereas G530 is oriented toward A1492. (**G**) The decoding center of Structure IV, with a closed 30S conformation, comprises A1493 and A1492 forming A-minor interactions with the first two base pairs of the codon-anticodon helix, whereas G530 is shifted toward helix 44 and interacts with A1492. This conformation resembles that of other 30S domain-closed structures in pre-accommodation-like 70S•EF-Tu•aa-tRNA complexes (*Stark et al., 2001*; *Valle, 2002*; *Schmeing et al., 2009*) and 70S complexes with fully accommodated A/A tRNA (*Voorhees et al., 2009*; *Jenner et al., 2010*; *Demeshkina et al., 2012*). Proteins are omitted for clarity in (**C–G**).

The following source data and figure supplements are available for figure 6:

**Source data 1.** Distances between Structures II, III and IV, reflecting the movement of the 30S shoulder domain from Structures II to III to IV, relative to the head and the body of the 30S subunit.
**Figure supplement 1.** Comparison of the 30S subunits of Structures II, III and IV reveals domain closure of the 30S subunit from Structure II to IV.
**Figure supplement 2.** The nucleotides at the decoding center and vicinity are resolved in the cryo-EM density.
**Figure supplement 3.** Conformational differences between the decoding centers of Structures I through IV.

(A1493 in the ON position). In summary, the key decoding center nucleotides in Structure II adopt the following conformations: A1493 ON, A1492 OFF, and G530 OFF.

In the decoding center of Structure III — corresponding to the intermediate state of domain closure — A1493 and A1492 contact the codon-anticodon helix, forming A-minor interactions with the first and second codon-anticodon base pairs (i.e., A1493 ON and A1492 ON; *Figure 6F*). G530 is in the *anti* conformation and shifted, along with the shoulder of the 30S subunit, toward A1492 (*Figures 6C,F* and *Figure 6—figure supplement 3C,G,J*). Here, G530 adopts a position between that in ribosomes with 'vacant' and 'filled' A sites (G530 SEMI-ON). The decoding center nucleotides in Structure III therefore adopt the conformations A1493 ON, A1492 ON, and G530 SEMI-ON.

Finally, the decoding-center nucleotides in Structure IV adopt conformations nearly identical (*Figures 6G* and *Figure 6—figure supplement 3D*, *H*, *K*) to those in the A-tRNA-bound ribosome (A1493/A1492/G530 ON) (*Selmer et al., 2006*; *Jenner et al., 2010*; *Demeshkina et al., 2012*). The 530 loop is shifted closer to A1493 and A1492, so that G530 interacts with A1492. This shift is coupled with the 30S subunit closure, also observed in 70S complexes with A-site tRNA (*Selmer et al., 2006*; *Jenner et al., 2010*; *Demeshkina et al., 2012*) and pre-accommodation-like 70S•EF-Tu•aa-tRNA complexes (*Stark et al., 2002*; *Valle, 2002*; *Schmeing et al., 2009*; *Fischer et al., 2015*).

## Structural mechanism of tRNA decoding in the A site

Our observation of the open and intermediate states at the decoding center in Structures II and III suggests how cognate tRNA is specifically selected during RelA activation. We propose the following structural mechanism of deacyl-tRNA decoding (*Video 1* and *Figure 7*). At early steps, interaction of the anticodon stem loop of a cognate or non-cognate tRNA occurs with the domain-open conformation of the 30S subunit, in which the decoding nucleotides are not positioned to stabilize the codon-anticodon helix. At this stage, the non-cognate tRNA dissociates prior to the closure of the 30S subunit, as the latter would require formation of the A-minor interactions by A1492 and

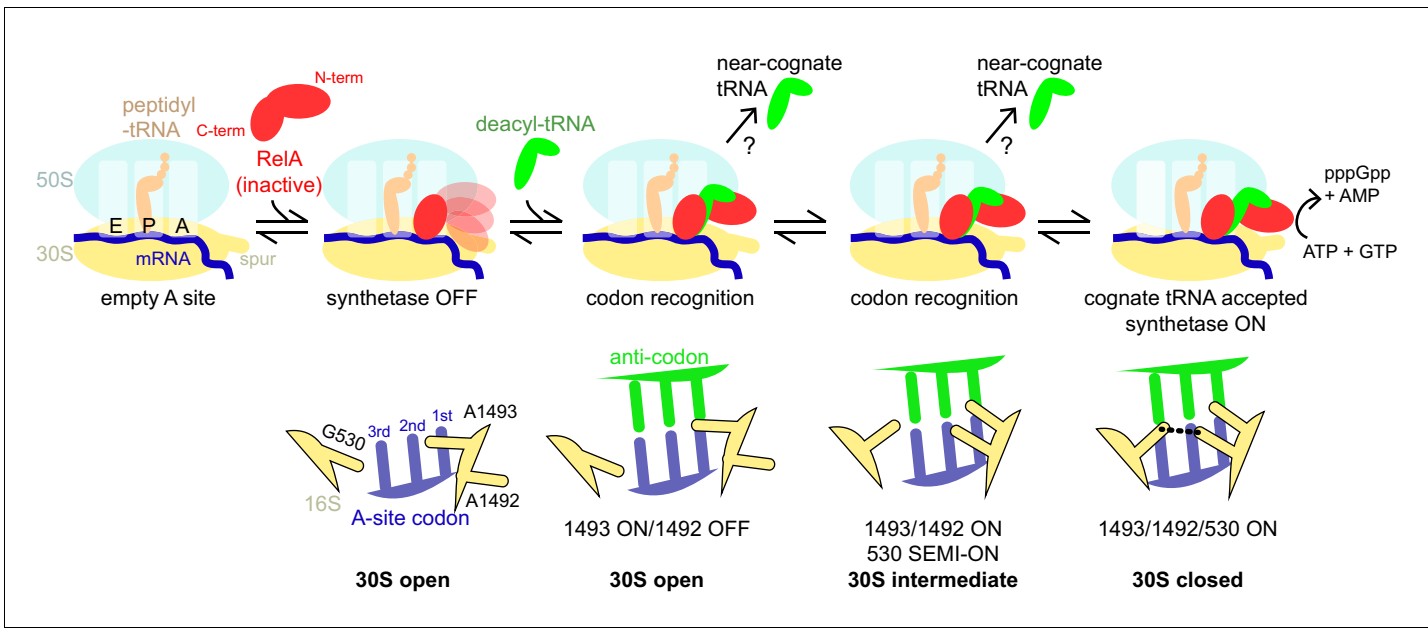

**Figure 7.** Schematic of the mechanism of RelA activation by the ribosome and cognate deacyl-tRNA. The C-terminal domains of RelA, the RIS and the ACT, bind the ribosome at the intersubunit bridge B1a near the vacant A site, but the synthetase remains unbound and inactive. When deacyl-tRNA binds to the ribosomal A site, the decoding center controls the selection of cognate tRNA, coupled with domain closure of the 30S subunit. The codon recognition checkpoints are mediated by distinct positions of the universally conserved nucleotides of the decoding center A1492, A1493 and G530. Upon binding of cognate deacyl-tRNA to the ribosome, the RelA synthetase domain is exposed in the vicinity of the 30S spur and is activated for (p) ppGpp synthesis by alleviation of RelA autoinhibition and interactions with the ribosome.

The following figure supplement is available for figure 7:

**Figure supplement 1.** Superpositions with structures of 70S-ribosome complexes suggest that RelA is displaced from ribosomes during tRNA accommodation and translocation.

A1493 with the Watson-Crick-paired codon-anticodon helix (*Ogle et al., 2001*; *Jenner et al., 2010*). In the case of cognate tRNA, acceptance of tRNA and 30S domain closure would be coupled with the formation of the A1492-G530 bridge (i.e., G530 ON) to stabilize the tRNA on the ribosome. The A1493-ON and/or A1493/A1492-ON states in Structures II and III may therefore serve as checkpoints on the path of acceptance of Watson-Crick base-paired tRNA and mRNA.

Our observation of previously elusive states of tRNA binding also suggests insights into the accuracy of aminoacyl-tRNA selection during elongation, a universally conserved mechanism responsible for the accurate transfer of genetic information. Despite a wealth of structural, biochemical and biophysical studies, the high-resolution structural understanding of this mechanism is limited and distinct mechanistic models have been proposed (e.g. [*Ogle et al., 2001*; *Demeshkina et al., 2012*]). Current structural understanding is limited to observation of the open and the closed conformations of the 30S subunit in the absence and presence of A-site tRNA, respectively (*Ogle et al., 2001*; *Jenner et al., 2010*). Biochemical (*Rodnina et al., 1994*; *Rodnina et al., 1995*; *Rodnina et al., 1996*; *Pape et al., 1998*) and biophysical (*Blanchard et al., 2004*; *Gonzalez et al., 2007*) studies demonstrated that tRNA accommodation includes an early short-lived intermediate – prior to forming the A/T state – during which non-cognate tRNAs can be rejected. However, early or intermediate states prior to the formation of the A/T state have not been structurally visualized at high resolution. Our observation of incompletely engaged tRNA in the domain-open or intermediate state of the 30S subunit suggests that similar states exist for the aminoacyl-tRNA•EF-Tu•GTP ternary complex. The tRNA acceptor arm positions are separated by ~5 Å in the early pre-accommodation step (Structure II) and the domain-closed step tRNA (Structure IV). This could be sufficient to keep EF-Tu farther from the GTPase-activating sarcin-ricin loop of 23S rRNA (*Voorhees et al., 2010*) at an early step, allowing non-cognate ternary complex dissociation prior to GTP hydrolysis, EF-Tu release and tRNA acceptance *via* domain closure. Further structural and biochemical studies of the pre-accommodating aminoacyl-tRNA•EF-Tu•GTP ternary complexes are required to test whether tRNA binding during the stringent response and during protein elongation are structurally similar.

## Model of RelA activation by the ribosome and cognate deacyl-tRNA

The stringent response must be rapidly initiated upon cellular stress (*Cashel and Gallant, 1969*), but under normal conditions the basal activity of RelA must remain low to avoid cell growth inhibition by (p)ppGpp (*Schreiber et al., 1991*). The cellular concentration of RelA under normal conditions is several orders of magnitude lower than that of ribosomes (*Pedersen and Kjeldgaard, 1977*; *Justesen et al., 1986*). Thus, a small number of cellular RelA molecules must use an efficient strategy to locate stress-activating ribosomes.

Could RelA remain bound to actively translating ribosomes or would it be displaced from these ribosomes until an activating stalled ribosome is found? During elongation, the ribosomal A site primarily interacts with the aminoacyl-tRNA, delivered by the EF-Tu•GTP•aa-tRNA ternary complex, and with elongation factor EF-G, which translocates peptidyl-tRNA from the A to the P site. Biochemical studies show that ribosomes bound with EF-Tu•GDPCP•Phe-tRNA[Phe] ternary complex can also bind RelA but do not result in (p)ppGpp synthesis (*Richter et al., 1975*; *Wendrich et al., 2002*). Superimposing *E. coli* 70S EF-Tu•GDP•Phe-tRNA[Phe]•kirromycin complex (*Fischer et al., 2015*) onto our 70S•RelA Structure I reveals no clashes between EF-Tu ternary complex and the RelA RIS and ACT domains (*Figure 7—figure supplement 1A*). The N-terminal domains are connected to the C-terminal domains with α-helical linkers, which might allow for the simultaneous binding of RelA and EF-Tu ternary complex. Steric hindrance with EF-Tu and the absence of interaction with deacyl-tRNA, however, would prevent activation of RelA, consistent with the biochemical data. Thus, the early steps of elongation are compatible with a ribosome-bound but inactive RelA.

Full accommodation and translocation of tRNA, however, would require displacement or relocation of RelA from bridge B1a, where the C-terminal superdomain is bound. When we superimpose Structure I and the structure of the 70S ribosome bound with three tRNAs (*Jenner et al., 2010*) (PDB: 3I8H), we observe a prominent steric clash between the ACT domain and the fully accommodated A-site tRNA (*Figure 7—figure supplement 1B*). Moreover, RelA does not bind pre-translocation and post-translocation 70S•EF-G complexes (*Wagner and Kurland, 1980*). Binding of EF-G to the pre-translocation ribosome stabilizes a rotated intersubunit state (*Frank and Agrawal, 2000*; *Cornish et al., 2008*; *Brilot et al., 2013*), in which bridge B1a is restructured and S19 is relocated by ~20 Å because of 30S subunit rotation (*Figure 7—figure supplement 1C*). This conformation would

disrupt the contact between the RIS domain and S19 and prevent binding of RelA to the rotated EF-G-bound ribosomes. In the non-rotated post-translation ribosome, wherein EF-G occupies the A site of the 30S subunit (*Frank and Agrawal, 2000*; *Gao et al., 2009*), steric hindrance between EF-G domain IV and Linker Helix 2 (*Figure 7—figure supplement 1D*) may prevent RelA from binding. Thus, in translating ribosomes, RelA cannot be activated and must be displaced from its binding site near bridge B1a.

Our structures suggest the following mechanism of RelA activation (*Figure 7*). The C-terminal domains of RelA associate with the ribosome near bridge B1a, as in Structure I. During amino-acid starvation, a cognate deacyl-tRNA binds the 30S A site as the codon-anticodon interaction is stabilized following step-wise rearrangements of the decoding center, as in Structures II, III and IV. At the other end of the tRNA, the deacylated CCA end pins the TGS domain against the body of the 30S subunit, exposing the dynamic synthetase domain near the spur. We propose that the activation of RelA synthetase is bifactorial. First, stabilization of RelA in the 70S•deacyl-tRNA complex alleviates the autoinhibition of the synthetase activity observed in ribosome-free RelA (*Schreiber et al., 1991*; *Gropp et al., 2001*; *Yang and Ishiguro, 2001a*). If the autoinhibition is due to inter-molecular inter-actions in oligomeric RelA (*Gropp et al., 2001*; *Yang and Ishiguro, 2001a*), as suggested by struc-tures of free dimeric ACT (*Eletsky et al., 2009*) and TGS domains (*Forouhar et al., 2009*), the dimerization surfaces of the ACT and TGS domains become disrupted by the ordering of the AH domain (*Figure 2D*) and interaction with the CCA end of the A/R tRNA (*Figure 3C*), respectively. Second, consistent with the observation of several conformations of the catalytic N-terminal domain, changes in the relative positions of the pseudo-hydrolase and synthetase domain, and/or in the interaction of the synthetase domains with the spur may contribute to the catalytic activation of the (p)ppGpp synthetase.

During translation under unstressed conditions, however, aminoacyl-tRNA accommodation in the A site and ribosome translocation displaces the C-terminus of RelA from the ribosome or at least from bridge B1a. We do not observe density for RelA when an A-site tRNA is fully accommodated, consistent with the model that RelA is displaced from the ribosome. Our structure-based mechanism is consistent with models in which RelA produces (p)ppGpp when bound to ribosomes with cognate deacylated tRNA, but is actively displaced from ribosomes during translation to limit (p)ppGpp pro-duction (*Elf and Ehrenberg, 2005*; *Li et al., 2015*). It is more difficult to reconcile our structures with the 'hopping' or 'extended hopping' models of RelA activation (*Wendrich et al., 2002*; *English et al., 2011*). The first model suggests that RelA is recruited to ribosomes with deacyl-tRNA already bound in the A site, activated to produce (p)ppGpp, and then passively dissociates to find another 70S•deacyl-tRNA complex (*Wendrich et al., 2002*). Although RelA binding to the 70S ribo-some following deacyl-tRNA is possible, our Structure I suggests that the presence of tRNA is not necessary for RelA binding, in agreement with biochemical observations (*Haseltine and Block, 1973*; *Ramagopal and Davis, 1974*; *Wagner and Kurland, 1980*; *Wendrich et al., 2002*). The 'extended hopping' model posits that activated RelA retains its activated state for some time after being released from 70S•deacyl-tRNA (*English et al., 2011*). Our structures, by contrast, indicate that RelA can bind a stalled (non-translating) ribosome before deacyl-tRNA arrives and that deacyl-tRNA binding is required to stabilize the extended RelA conformation.

Our observation of several conformations of stringent response complexes raises the question whether one or more A/R-tRNA–bound states activate RelA. RelA might be prepared for catalysis in all three states. Alternatively, continuous transitions between Structures II, III, and IV could be required for synthesis of (p)ppGpp, for example, by distributing the alignment of catalytic residues, substrate binding and positioning, and product release between these distinct states. A third model is that RelA is activated by only one state. We favor mechanisms, in which Structure IV is required to activate RelA, either as part of an 'activating' ensemble or as a sole activating complex (*Figure 7*). Near-cognate tRNAs do not trigger (p)ppGpp synthesis (*Haseltine and Block, 1973*). Our tRNA accommodation model predicts that near-cognate tRNAs would sample conformations similar to those in Structures II or III, but do not proceed to the domain-closed 'acceptance' state (Structure IV). Thus, we propose that Structure IV, with an accommodated cognate tRNA anticodon stem loop and closed decoding center, is necessary to activate RelA. Notably, the space in which the N-termi-nal domains are located between the sarcin-ricin loop and the spur is more constricted in the domain-closed state, highlighting the possible role of interactions between these domains and ribo-somal RNA in (p)ppGpp-synthetase activation. After submission of this manuscript, two studies

reported RelA-bound structures of the 70S ribosome in the presence of a non-hydrolyzable ATP analog and GTP (*Arenz et al., 2016*; *Brown et al., 2016*). As such, the published complexes describe substrate-bound states of RelA, whereas our complex lacks ATP and GTP and, therefore, describes RelA states prior to substrate binding. In both published studies, a single predominant global structure is reported, however the conformational variability in the 30S domains (*Brown et al., 2016*) and N-terminal domains of RelA (*Arenz et al., 2016*; *Brown et al., 2016*) is noted, consistent with our observations. The predominance of the single ribosome conformation is likely due to the use of the antibiotic paromomycin in one study (*Brown et al., 2016*), which stabilizes a domain-closed 30S conformation (*Ogle et al., 2001*) similar to our Structure IV, or to the use of RelA substrate analogs, or to differences in cryo-EM dataset sizes or in classification procedures. Further work will address the roles of 30S inter-domain rearrangements, RelA inter-domain rearrangements and interactions between the synthetase domain and the 30S spur in the activation of RelA.

## Materials and methods

### Ribosome•RelA complex preparation

*E.coli* RelA coding sequence was obtained from ASKA Clone(-) library (National BioResource Project, NIG, Japan) and was subcloned into the expression vector pET24b to carry an N-terminal 6xHis-tag. RelA was overexpressed and purified essentially as described (*Knutsson Jenvert and Holmberg Schiavone, 2005*; *Agirrezabala et al., 2013*). The crude *E. coli* RelA-containing lysate was passed through nickel resin (HisPur Ni-NTA Resin, Thermo Fisher Scientific) and washed out with elution buffer (20 mM K-Hepes (pH 7.5), 1000 mM KCl, 1 mM $MgCl_2$, 250 mM imidazole, 15% glycerol and 6 mM β-mercaptoethanol, freshly added in this and subsequent steps). The elution product was dialyzed against a low-salt buffer (10 mM Tris-HCl (pH 8), 60 mM KOAc, 14 mM Mg(OAc)$_2$, 0.5 mM EDTA, 15% glycerol and 5 mM β-mercaptoethanol) to precipitate out RelA. RelA was re-dissolved in the storage buffer (20 mM K-Hepes (pH 7.5), 1000 mM KCl, 1 mM $MgCl_2$, 15% glycerol and 5 mM βME). The purity of the recovered protein (>95%) was confirmed by SDS-PAGE analysis. 70S ribosomes were prepared from MRE600 *E. coli* essentially as described (*Moazed and Noller, 1986*, *1989*) and stored in the ribosome-storage buffer (100 mM Tris-HCl (pH 7.0), 100 mM $NH_4Cl$, 10.5 mM $MgCl_2$, 0.5 mM EDTA, 5 mM β-mercaptoethanol) at −80°C. tRNA$^{fMet}$ and tRNA$^{Phe}$ were purchased from ChemBlock. RNA, containing the Shine-Dalgarno sequence and a linker to place the AUG codon in P site and UUC codon in the A site (GGC AAG GAG GUA AAA AUG UUC AAA AAA), was synthesized by IDT DNA.

The 70S•RelA•mRNA•P-tRNA$^{fMet}$•A/R-tRNA$^{Phe}$ complex was prepared as follows. 4 μM 70S ribosomes were incubated with 20 μM mRNA, 8 μM tRNA$^{fMet}$ and 8 μM tRNA$^{Phe}$ (all final concentrations) for 30 min at 37°C, in Buffer A (20 mM Hepes-KOH pH 7.4, 120 mM KCl, 6 mM $MgCl_2$, 2 mM spermidine, 0.05 mM Spermine, 6 mM β-mercaptoethanol). RelA was then added at 5 μM (final concentration) and the solution was incubated for 30 min at 37°C. The complex was diluted in Buffer A and supplemented with tRNA$^{Phe}$ and RelA to the following final concentrations: 40 nM 70S, 200 nM mRNA, 80 nM tRNA$^{fMet}$, and 1 μM tRNA$^{Phe}$ and 2 μM RelA. This diluted reaction was allowed to equilibrate at least 5 min at 37°C prior to application on cryo grids.

### Grid preparation

Holey-carbon grids (C-flat 1.2–1.3, Protochips) were coated with a thin layer of carbon and glow discharged at 20 mA with a negative polarity setting for 45 s in an EMITECH K100X glow discharge unit. 2 μL of the diluted sample was applied to the grids. After a 10-second incubation, the grids were blotted for 4 s and plunged into liquid ethane using a CP3 cryo plunger (Gatan Inc.) at room temperature and ~75% humidity.

### Electron microscopy

A dataset of 564,385 particles was collected as follows. 2992 and 5211 movies were automatically collected using SerialEM (*Mastronarde, 2005*) in two sessions on a Titan Krios electron microscope (FEI) operating at 300 kV and equipped with K2 Summit direct electron detector (Gatan Inc.) using 0.5 to 2.2 μm underfocus. 25 frames per movie were collected over 10 s at 4 e⁻/Å²/s for a total dose of 40 e⁻/Å² on the sample. The super-resolution pixel size was 0.82 Å on the sample.

## Image processing

Particles were extracted from aligned movie sums as follows. Movies were processed using IMOD (*Kremer et al., 1996*) to decompress frames and apply the gain reference. Movies were drift-corrected and exposure-filtered using unblur (*Grant and Grigorieff, 2015b*). Magnification anisotropy of the movie sums was corrected with mag_distortion_estimate and mag_distortion_correct (*Grant and Grigorieff, 2015a*). CTFFIND3 (*Mindell and Grigorieff, 2003*) was used to determine defocus values. 2233 movies from the first dataset and 207 movies from the second dataset with high drift, low signal, heavy ice contamination, or very thin ice were excluded from further analysis after inspection of image sums and power spectra from CTFFIND3. Particles were automatically picked from 10x binned images using Signature (*Chen and Grigorieff, 2007*) with a ribosome reference (18 representative reprojections of EM databank map 1003 (*Gabashvili et al., 2000*), which was low-pass filtered to 50 Å). 480x480 pixel boxes with particles were extracted from super-resolution images, and the stack and FREALIGN parameter file were assembled in IMAGIC (*van Heel et al., 1996*). To speed up processing, 2x, 4x, and 6x binned image stacks were prepared using resample.exe, which is part of the FREALIGN distribution.

FREALIGN v9 (versions 9.07–9.11) was used for all steps of refinement and reconstruction (*Lyumkis et al., 2013*) (*Figure 1—figure supplement 1*). The 6x binned image stack was initially aligned to a ribosome reference (EM databank map 1003, [*Gabashvili et al., 2000*]) using five rounds of mode 3 (global search) alignment including data in the resolution range from 300 Å to 30 Å. Next, the 2x binned, and later the unbinned image stacks were successively aligned against the common reference using mode 1 (local refinement) including data up to a high-resolution limit of 6 Å whereupon the resolution of the common reference stopped improving (FSC (0.143) = 3.5 Å). Subsequently, the refined parameters were used for classification of the 6x binned stack into 5–25 classes in 30–80 rounds using resolutions from 12 to 300 Å. This yielded multiple RelA-containing classes, one of which we used to build an initial atomic model. We found that using a three-dimensional (3D) mask (described below) improved the separation of the RelA bound classes during classification. In the final classification with the 3D mask, the 4x binned stack was separated into 15 classes in 50 rounds that included data between 8 to 300 Å resolution. The 3D mask was created using Spider (*Frank et al., 1996*) by generating a density map, low-pass filtered to 30 Å, from our initial atomic model and including the following components: RelA, A/R-, P- and E-site tRNAs, and most of the 30S subunit (a 10-Å sphere around protein S2 was excluded because S2 appeared substoichmetric or disordered in the complex). The mask was applied to reference volumes in Frealign such that parts of the ribosome outside of the mask were low-pass filtered to 30 Å (*Grigorieff, 2016*). A five-pixel cosine edge was used on the mask and the masking filter function. This final classification revealed seven high-resolution classes and eight junk classes (noisy or low-resolution). The high-resolution classes differed in tRNA and RelA occupancies and 30S conformations (*Figure 1—figure supplement 1*). For the classes bound with RelA (Structures I - IV), particles with > 50% occupancy were extracted from the 1x binned stack, and the four final maps were prepared following three rounds of mode 1 refinement to 8 Å resolution. To aid model building of RelA domains, we performed local refinements within 3D spherical masks. The particles belonging to Structures II, III and IV were combined and masks encompassing either the C-terminal domains (RIS and ACT) or the AH domain were applied to reference volumes in FREALIGN, so that parts of the ribosome outside of the mask were downweighted to 10% density during 10 rounds of mode 1 refinement to 8 Å resolution (*Figure 2—figure supplement 1C–D*). Finally, to resolve the N-terminal regions of RelA, we subclassified Structures II, III and IV individually, using a focus mask (a sphere, 80 Å in diameter) that encompassed the pseudo-hydrolase and synthetase density (*Figure 4—figure supplement 1*). 100 rounds of classification were run, separating particles into 3, 4, 5 or 7 classes and using data between either 12 to 300 or 20 to 300 Å resolution.

The maps used for structure refinements were B-factor sharpened using B-factors of -50 to -200 using bfactor.exe (included with the FREALIGN distribution [*Lyumkis et al., 2013*]). FSC curves were calculated by FREALIGN for even and odd particle half-sets (*Figure 1—figure supplement 2*).

## Model building and refinement

Recently reported high-resolution cryo-EM structure of the 70S•EF-Tu•aa-tRNA complex (PDB: 5AFI) (*Fischer et al., 2015*), excluding EF-Tu and P- and E-site tRNAs, was used as a starting model for

structure refinement. The starting structural models for tRNA^fMet in the P and E sites were adopted from the 70S•RF2•tRNA crystal structure (*Korostelev et al., 2008*). The starting model for RelA was created by homology modeling and *de novo* modeling. The TGS domain of CLOLEP_03100 from *Clostridium leptum* (PDB: 3HVZ; [*Forouhar et al., 2009*]) and the nuclear magnetic resonance structure of the ACT domain of GTP pyrophosphokinase from *Chlorobium tepidum* (PDB: 2KO1 [*Eletsky et al., 2009*]) were used for homology modeling employing SWISS-PROT (*Bairoch et al., 2004*). *De novo* structure prediction by ROSETTA (*Kim et al., 2004*) and Quark (*Xu and Zhang, 2012*) was used to build the RIS domain, for which no homologous structures were found by sequence homology. Our initial modeling of the RIS domain revealed the zing-finger fold, according to DALI server (*Holm and Rosenström, 2010*), however unambiguous assignment of some amino-acid side chains was challenging. In our final refinements, we adopted the RIS domain from the recently published structure of *E. coli* RelA (*Brown et al., 2016*), in which most side-chain positions agree with our densities. The initial model for the AH domain was obtained using I-TASSER (*Yang et al., 2015*). Cryo-EM densities, obtained using a spherical mask around the N-terminal RelA domains, suggest a helical region between the synthetase and TGS domains (aa 360–380), consistent with the similarly-positioned long helix in the recently determined structures of the homologous small alarmone synthetase 1 (SAS1; [*Steinchen et al., 2015*]). The homology model for the N-terminal region (residues 16–351), obtained using the crystal structure of RelSeq (PDB: 1VJ7) (*Hogg et al., 2004*), was fitted using Chimera (*Pettersen et al., 2004*) as a single rigid group into the low-resolution maps obtained by sub-classification of Structures II, III and IV, as shown in *Figure 4—figure supplement 1*. The linkers between the domains and parts of the domains, whose amino acid side chain positions could not be unambiguously determined from homology modeling and density maps, were modeled as poly-alanine.

Structures I-IV were refined by real-space simulated-annealing refinement (*Chapman, 1995*; *Korostelev et al., 2002*) against corresponding maps, excluding the central domains (Structure I) and the N-terminal domains (Structures I-IV). Atomic electron scattering factors, obtained from Dr. Tamir Gonen (*Gonen et al., 2005*), were used during refinement. Refinement parameters, such as the relative weighting of stereochemical restraints and the experimental energy term, were optimized to produce the optimal structure stereochemistry, real-space correlation coefficient and R-factor, which report on the fit of the model to the map (*Zhou et al., 1998*). Secondary-structure restraints, comprising hydrogen-bonding restraints for ribosomal proteins and base-pairing restraints for RNA molecules were employed as described (*Laurberg et al., 2008*). The resulting structural models have good stereochemical parameters, characterized by low deviation from ideal bond lengths and angles (*Figure 1—source data 1*).

Figures were prepared in Chimera and Pymol (*DeLano, 2002*).

## Acknowledgements

We thank Cha San Koh for assistance with RelA and ribosome purification; Chen Xu and Mike Rigney for help at the cryo-EM facility at Brandeis University; Zhiheng Yu, Jason de la Cruz, and Chuan Hong for data collection at Janelia Research Campus; Darryl Conte Jr. for assistance with manuscript preparation; Dmitri Ermolenko, Alexei V Korennykh and members of the Grigorieff and Korostelev laboratories for helpful comments on the manuscript. This study was supported by NIH Grants R01 GM106105 (to AAK) and P01 GM62580 (to NG). ABL is a Howard Hughes Medical Institute Fellow of the Helen Hay Whitney Foundation.

## Additional information

### Competing interests

NG: Reviewing editor, e*Life*. The other authors declare that no competing interests exist.

### Funding

| Funder | Grant reference number | Author |
| --- | --- | --- |
| Howard Hughes Medical Institute | | Nikolaus Grigorieff |

| National Institutes of Health | RO1 GM106105 | Andrei A Korostelev |
| National Institutes of Health | PO1 GM62580 | Nikolaus Grigorieff |
| Helen Hay Whitney Foundation | | Anna B Loveland |

The funders had no role in study design, data collection and interpretation, or the decision to submit the work for publication.

### Author contributions

ABL, Wrote the manuscript, Collected and analyzed cryo-EM data, Contributed to manuscript finalization, Analysis and interpretation of data; EB, Prepared 70S•RelA complexes, Acquisition of data, Contributed unpublished essential data or reagents, Contributed to manuscript finalization; RM, Subcloned the relA gene and assisted in 70S•RelA complex preparation, Contributed unpublished essential data or reagents, Contributed to manuscript finalization; YZ, Prepared 70S•RelA complexes, Contributed unpublished essential data or reagents, Contributed to manuscript finalization; AFB, Assisted with cryo-EM data collection and analyses, Contributed to manuscript finalization; NG, Oversaw cryo-EM data processing, Designed the project, Wrote the manuscript, Analysis and interpretation of data, Contributed to manuscript finalization; AAK, Built and refined structural models, Wrote the manuscript, Designed the project, Analysis and interpretation of data, Contributed to manuscript finalization

### Author ORCIDs

Nikolaus Grigorieff, http://orcid.org/0000-0002-1506-909X
Andrei A Korostelev, http://orcid.org/0000-0003-1588-717X

# Additional files

## Major datasets

The following datasets were generated:

| Author(s) | Year | Dataset title | Dataset URL | Database, license, and accessibility information |
|---|---|---|---|---|
| Loveland AB, Bah E, Madireddy R, Zhang Y, Brilot AF, Grigorieff N, Korostelev AA | 2016 | Structure of RelA bound to ribosome in absence of A/R tRNA (Structure I) | http://www.rcsb.org/pdb/explore/explore.do?structureId=5KPS | Publicly available at the RCSB Protein Data Bank (accession no: 5KPS) |
| Loveland AB, Bah E, Madireddy R, Zhang Y, Brilot AF, Grigorieff N, Korostelev AA | 2016 | Structure of RelA bound to ribosome in presence of A/R tRNA (Structure II) | http://www.rcsb.org/pdb/explore/explore.do?structureId=5KPV | Publicly available at the RCSB Protein Data Bank (accession no: 5KPV) |
| Loveland AB, Bah E, Madireddy R, Zhang Y, Brilot AF, Grigorieff N, Korostelev AA | 2016 | Structure of RelA bound to ribosome in presence of A/R tRNA (Structure III) | http://www.rcsb.org/pdb/explore/explore.do?structureId=5KPW | Publicly available at the RCSB Protein Data Bank (accession no: 5KPW) |
| Loveland AB, Bah E, Madireddy R, Zhang Y, Brilot AF, Grigorieff N, Korostelev AA | 2016 | Structure of RelA bound to ribosome in presence of A/R tRNA (Structure IV) | http://www.rcsb.org/pdb/explore/explore.do?structureId=5KPX | Publicly available at the RCSB Protein Data Bank (accession no: 5KPX) |
| Loveland AB, Bah E, Madireddy R, Zhang Y, Brilot AF, Grigorieff N, Korostelev AA | 2016 | RelA bound to 70S ribosome in absence of A/R tRNA (Structure I) | http://emsearch.rutgers.edu/atlas/8279_summary.html | Publicly available at the EMDataBank (accession no: 8279) |

| Loveland AB, Bah E, Madireddy R, Zhang Y, Brilot AF, Grigorieff N, Korostelev AA | 2016 | RelA bound to 70S ribosome in presence of A/R tRNA (Structure II) | http://emsearch.rutgers.edu/atlas/8280_summary.html | Publicly available at the EMDataBank (accession no: 8280) |
|---|---|---|---|---|
| Loveland AB, Bah E, Madireddy R, Zhang Y, Brilot AF, Grigorieff N, Korostelev AA | 2016 | RelA bound to 70S ribosome in presence of A/R tRNA (Structure III) | http://emsearch.rutgers.edu/atlas/8281_summary.html | Publicly available at the EMDataBank (accession no: 8281) |
| Loveland AB, Bah E, Madireddy R, Zhang Y, Brilot AF, Grigorieff N, Korostelev AA | 2016 | RelA bound to 70S ribosome in presence of A/R tRNA (Structure IV) | http://emsearch.rutgers.edu/atlas/8282_summary.html | Publicly available at the EMDataBank (accession no: 8282) |

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
