## [Decision Letter]

Thank you for submitting your article "Ribosome•RelA structures reveal the mechanism of stringent response activation" for consideration by *eLife*. Your article has been favorably evaluated by Wenhui Li (Senior editor) and three reviewers, one of whom is a member of our Board of Reviewing Editors. The reviewers have opted to remain anonymous.

The reviewers have discussed the reviews with one another and the Reviewing Editor has drafted this decision to help you prepare a revised submission.

All reviewers find the submitted manuscript detailing the interaction of RelA with the bacterial ribosome to be of interest to experts in the field as well as a general audience. Importantly, the multiple states of bound RelA that are captured in the study provide interesting texture that allow the authors to discuss potential intermediate states in activation of the stringent response (as well as potentially in tRNA accommodation during decoding).

The reviewers support publication in *eLife* after several issues have been addressed.

First, the study now needs to acknowledge the recent publication by Ramakrishnan and colleagues in Nature and how the present data extend and fit into this new context.

Second, the authors need to more carefully discuss why they think they observe intermediate states that have not been observed by others and whether the resolution of the current structures is sufficient to define, for example, the conformations of nucleobases in the 16S rRNA.

Finally, the reviewers felt that comparisons to the innate immune sensors OAS1 and cGas1 were overstated and should be reserved to a much more modest section in the Discussion.

---

## [Author Response]

*First, the study now needs to acknowledge the recent publication by Ramakrishnan and colleagues in Nature and how the present data extend and fit into this new context.*

After the submission of our manuscript, two manuscripts (Brown et al., Nature; Arenz et al., Nucleic Acid Research) reported cryo-EM structures of 70S*RelA complexes bound with deacyl-tRNA in the 30S A site. Each study, performed in the presence of RelA substrate analogs, describes a structure that appears similar to our Structure IV (our dataset was obtained in the absence of RelA substrate analogs). We note that those structures agree with our initial proposal that our Structure IV may represent an activated form of RelA and we now discuss these publications in the context of our manuscript.

*Second, the authors need to more carefully discuss why they think they observe intermediate states that have not been observed by others and whether the resolution of the current structures is sufficient to define, for example, the conformations of nucleobases in the 16S rRNA.*

In the publications by Brown et al. and Arenz et al., conformational heterogeneity of the 30S subunit and RelA is noted that appears to correspond with the conformational changes that we observe between Structures II and IV. Our work is therefore consistent with these studies. It is possible that the other studies discuss only one predominant state due to the use of the antibiotic paromomycin, which stabilizes a state similar to that corresponding to our Structure IV (as in Brown et al). It is also possible that the smaller size of the dataset (as in Arenz et al) did not allow further classification of the different states of RelA, or because the authors simply chose not to analyze the heterogeneity they noticed.

We have added figures showing that individual nucleobases can be resolved in the ribosome core of 16S rRNA, and we note that further studies are necessary to reveal the role of detailed conformational rearrangements in tRNA decoding and RelA activation.

*Finally, the reviewers felt that comparisons to the innate immune sensors OAS1 and cGas1 were overstated and should be reserved to a much more modest section in the Discussion.*

We have substantially reduced our discussion of the possible link between RelA and innate immune sensors to three sentences.